# Phosphatidylcholine Passes by Paracellular Transport to the Apical Side of the Polarized Biliary Tumor Cell Line Mz-ChA-1

**DOI:** 10.3390/ijms20164034

**Published:** 2019-08-19

**Authors:** Wolfgang Stremmel, Simone Staffer, Ralf Weiskirchen

**Affiliations:** 1Institute of Pharmacy and Molecular Biotechnology, University of Heidelberg, D-69120 Heidelberg, Germany; 2University Clinics of Heidelberg, D-69120 Heidelberg, Germany; 3Institute of Molecular Pathobiochemistry, Experimental Gene Therapy and Clinical Chemistry, RWTH University Hospital Aachen, D-52074 Aachen, Germany

**Keywords:** Mz-ChA-1 cells, biliary epithelial cells, phosphatidylcholine, mucus, tight junctions, paracellular transport

## Abstract

Phosphatidylcholine (PC) translocation into mucus of the intestine was shown to occur via a paracellular transport across the apical/lateral tight junction (TJ) barrier. In case this could also be operative in biliary epithelial cells, this may have implication for the pathogenesis of primary sclerosing cholangitis (PSC). We here evaluated the transport of PC across polarized cholangiocytes. Therefore, the biliary tumor cell line Mz-ChA-1 was grown to confluency. In transwell culture systems the translocation of PC to the apical compartment was analyzed. After 21 days in culture, polarized Mz-ChA-1 cells revealed a predominant apical translocation of choline containing phospholipids including PC with minimal intracellular accumulation. Transport was suppressed by TJ destruction employing chemical inhibitors and pretreatment with siRNA to TJ forming proteins as well as the apical transmembrane mucin 3 as PC acceptor. Apical translocation was dependent on a negative apical electrical potential created by the cystic fibrosis transmembrane conductance regulator (CFTR) and the anion exchange protein 2 (AE2). It was stimulated by apical application of secretory mucins. The results indicated the existence of a paracellular PC passage across apical/lateral TJ of the polarized biliary epithelial tumor cell line Mz-ChA-1. This has implication for the generation of a protective mucus barrier in the biliary tree.

## 1. Introduction

It has been known for a long time that tight junctions (TJ) are responsible for sealing epithelial cells at their apical–lateral side. This enables stable boundaries with their respective functional implication, e.g., at the blood–brain barrier. In intestinal mucosa, they prevent the attack of microbiota. Additionally, they serve the environmental control by allowing water and electrolyte exchange. Macromolecules were not known to pass this barrier. However, in recent studies the novel pathway of phosphatidylcholine (PC) transport across lateral TJ to the luminal side of mucosal cells was described [1].

After apical translocation, PC from systemic sources (lipoproteins) is enriched within the mucus by binding to mucin 2 to establish a hydrophobic barrier against the colonic lumen. Indeed, genetic mouse models with intestinal specific TJ deletion revealed an impaired mucus barrier with lack of PC [2]. This caused an ulcerative colitis (UC) phenotype [2]. The intrinsic lack of mucus PC, the disposition for microbiota invasion and the consequent inflammation matches the situation in human UC [3,4]. The disease is often associated with primary sclerosing cholangitis (PSC), the pathogenesis of which remains obscure [5]. It was indeed postulated that the luminal lack of PC may be of etiological relevance. When canalicular secretion of PC through the multiple drug resistance gene 2 (MDR2) was deleted in genetically modified mice, a severe cholangitis occurred [6,7]. It is believed that PC is secreted to neutralize the aggressiveness of bile acids by packing them into micelles. Thus, it prevents the attack of bile acids against the plasma membrane phospholipid bilayer of biliary epithelial cells. However, analysis in patients with PSC did not show a lack of PC in bile [8].

These studies neglected the fact that between lumen and the epithelial cells, there is a mucus layer containing secretory mucins. They bind PC to constitute a protective shield towards bile. It is unlikely that mucus PC originates from bile, containing PC-bile acid micelles with high mutual affinity. Therefore, we assume that the biliary mucus compartment is fed with PC from systemic sources as it is the case in intestinal mucosa [2]. We further assume that PC passes through lateral TJ to the apical side of the biliary epithelial cells. Indeed, a common genetic defect in intestinal and biliary epithelial cells could explain the high association of UC and PSC. To prove this concept, we first analyzed the pathway of PC from basal to the apical side employing the biliary tumor cell line Mz-ChA-1. These cells share the characteristics of physiologic biliary epithelial cells [9].

## 2. Results

According to our hypothesis, the transport of PC to the apical side of biliary epithelial cells requires the presence of TJ. We utilized the well-characterized biliary epithelial tumor cell line Mz-ChA-1 [9] for our experiments and grew them in transwell tissue culture systems for up to 21 days. This cell line has a highly differentiated phenotype producing large quantities of mucus. In comparison to many other human biliary tract carcinoma cell lines such as TGBC-1, TBC-51, Mz-ChA-2 and SK-ChA-1, the cell line Mz-ChA-1 is 10–1000 times less invasive through both the collagen and the basement membrane [10]. The cell line Mz-ChA-1 was originally described as an adenocarcinoma cell line. However, many subsequent studies showed that the cell line has many features of human cholangiocyte cells including TJ complexes rendering these cells as appropriate tool for respective studies [11,12,13,14].

While at day three only marginal transepithelial resistance (TER) was detectable with 120 ± 40 Ω (non-polarized cells), it increased to 220 ± 30 Ω at day nine and remained stable after 21 days (polarized cells) with 400 ± 80 Ω. Western blot analysis of representative TJ proteins confirmed their presence after 21 days in culture (Figure 1).

Then apical and basal PC transport was evaluated by a respective translocation of PC, provided to the opposite compartment together with taurocholate (TC) at indicated concentrations. For comparison inulin as a non-cell permeable compound was analyzed. Indeed, polarized Mz-ChA-1 cells had the capacity to translocate PC to the apical compartment, whereas basal transport diminished (Figure 2). Inulin, instead, revealed preferential transport to the basal side. These observations were in contrast to non-polarized Mz-ChA-1 cells which resulted in equilibrated PC and inulin concentrations between apical and basal compartments (Figure 2).

The intracellular accumulation was tested in non-polarized cells incubated with 100 μM for 1 h and accounted for <5% of the incubated PC, whether it was provided together with TC or albumin. Only when PC was provided with apolipoprotein B (ApoB) an uptake rate of 12.84 ± 5 nmol·mg protein^−1^·h^−1^ was observed indicative of lipoprotein-mediated endocytosis of PC. In comparison, 1 h uptake rates for radiolabeled TC and oleate, the later complexed with TC or albumin, were significantly higher (*p* < 0.01) (Figure 3).

Apical transport of PC increased linearly with time of exposure and incubated PC concentrations, was temperature dependent with the highest rates between 25–37 °C, and a pH optimum between pH 7.0 to pH 8.0 (Figure 4).

Transport was specific for the choline containing phospholipids PC, lysophosphatidylcholine (LPC) and sphingomyelin (SM) but not for other phospholipids (Figure 5).

Apical transport was inhibited when a positive charge was generated by application of ammonium chloride (NH_4_Cl), sodium thiocyanate (NaSCN) or urea, but remained stable by application of negative charge to the apical surface (Figure 6) [1]. Negative charge in vivo is generated by the cystic fibrosis transmembrane conductance regulator (CFTR) or the anion exchange protein 2 (AE2) which are both present in cholangiocytes [15].

When CFTR or AE2 were reduced by siRNA pretreatment, PC transport was strongly inhibited (Figure 7).

The hypothesis of a TJ-mediated translocation process was supported by the observation of its inhibition by exposure of polarized Mz-ChA-1 cells to acetaldehyde (ACA) vapor or the selective peroxisome proliferator-activated receptor γ (PPARγ) inhibitors T0070907 or GW9962 (Figure 8).

Moreover, apical transport was diminished by pretreatment of polarized Mz-ChA-1 cells with siRNAs against members of the TJ complex or their upstream control proteins kindlin-1 and -2 (Figure 7). Although cholangiocytes do not synthesize mucin 1 and 2, they were shown to have mucin 3 which is a transmembrane protein serving in accepting apically transported PC as it was shown in two previous studies [1,2]. Its inhibition by siRNA reduced PC translocation significantly. From mucin 3 PC is handled to secreted mucin 1 and 2 which originates from goblet cells [16]. When it was applied to the apical surface of polarized Mz-ChA-1 cells transport was dose dependently enhanced as it was also the case when increasing concentrations of TC were apically added which favor the drainage of transported PC to the apical side by incorporation of PC into micelles (Figure 9). However, it has to be considered that Mz-ChA-1 cells as well as biliary epithelial cells in vivo do not exhibit a mucin 2 containing mucus layer.

To document a vectoral transport of PC from the basal to the apical surface, it was tested when increasing, but equal concentrations of PC in both compartments were provided. In contrast to inulin, where basal transport was higher than apical translocation, PC was overwhelmingly transported apically with only marginal appearance at the basal side indicating a predominant paracellular transport across TJ to the luminal surface (Figure 10).

## 3. Discussion

Kinetic analyses of the translocation of PC across the polarized biliary epithelial tumor cell line Mz-ChA-1 revealed a TJ-mediated process. This was as unexpected as it was for TJ-mediated PC transport in the intestinal mucosa [1,2]. This transport is strongly unidirectional to the apical side, driven by a negative apical charge generated by CFTR and AE2. The translocated PC is bound to mucin 3 from where it is handled to mucin 1 and 2 in the mucus layer of the biliary tract.

The PC in mucus serves in the biliary tree to protect the epithelial cells from the attack of bile acids, which achieve concentrations in the millimolar range within the small ducts [17]. This can be neutralized by canalicular secretion of PC via MDR2/MDR3 (ABCB4) through the formation of micelles.

However, the excessive load of bile acids could in vivo take PC from the mucus layer still maintaining a shield towards biliary epithelium, because there is a constant basal supply to the apical surface. In fact, as shown in this study, PC secretion is proportionally increased with TC in the apical (biliary) compartment. Thus, PC in cell membranes is not exposed to the luminal bile acid load. However, when, for example by a genetic defect, the apical PC secretion through the biliary epithelial cell layer is impaired, they are attacked, and a cholangitis occurs. This hypothesis will now be proven in a genetic mouse model where the biliary TJ are deleted. The observation of a disturbed TJ barrier as a potential cause of impaired secretion of PC to biliary mucus could shed light on the pathogenesis of PSC. It associates to UC with a genetically determined disruption of the TJ barrier and diminished secretion of PC to the intestinal mucus [2,3,4]. When this relation can also be verified in the biliary epithelium, new therapeutic strategies for PSC will be developed.

The shortcoming of this study is the use of a tissue culture model, even with a cholangiocyte-derived tumor cell line. However, biliary epithelial cells are not obtainable in high yield, in reproducible fashion and sufficient functionality. Moreover, isolated cholangiocytes do not exhibit a mucus layer. It can only be pointed out that the employed Mz-ChA-1 cell line exhibits TJ complexes and is more differentiated in comparison to other cell lines obtained from cholangiocellular carcinomas. Another shortcoming is the lack of data on the structure of the TJ under the various experimental conditions. Moreover, imaging of transport employing fluorescent PC could not be examined in these experiments. All of these shortcomings can be tackled, when a genetic mouse model with TJ deletion selective for the biliary epithelium becomes available.

## 4. Materials and Methods

### 4.1. Cell Culture Transport Studies

Transwell tissue culture for basal–apical polarization was established for the biliary epithelial tumor cell line Mz-ChA-1 obtained from Dr. Alexander Knuth (Mainz, Germany) in our laboratory [10]. In each well of the 12-well collagen-coated transwell culture dish 7.5 × 10^4^ cells (corresponding to 80 ± 18 μg protein) were placed and cultured in Dulbecco’s modified Eagle’s medium containing 5% fetal calf serum ((FCS) (Life Technologies, Carlsbad, CA, USA). They were grown for up to 21 days and polarity was examined by TER [1].

Translocation of radiolabeled substrates to the apical or basal side was examined by its application to one side and its recovery in the opposite compartment. For equilibrium distribution studies substrates were located in both compartments and transport to apical and basal side was determined. Standard incubation procedures used 10 mM PC (100,000 cpm) bound to 10 mM TC in phosphate buffered saline (PBS) (pH 7.4) at 37 °C in 1 mL placed at basal side and recovery was analyzed after 1 h in the upper compartment with 1 mL 10 mM TC–PBS. As substrates we used [^3^H]phosphatidylcholine (PC), [^14^C]lysophosphatidylcholine (LPC), [^3^H]sphingomyelin (SM), [^14^C]phosphatidylethanolamine (PE), [^3^H]phosphatidylinositol (PI), [^3^H]palmitate (PA), [^3^H]oleate (OA), [^3^H]taurocholate (TC) and [^14^C]inulin (all brought up with unlabeled substrate to desired concentrations). Transport characteristics included dependency on time, concentration, temperature and pH and was evaluated as described [1]. For driving force analysis, the 10 mM TC in the apical medium was applied in different buffers at 130 mM and pH 7.0 which generated a more positive charge in the medium (NH_4_Cl, Na-thiocyanate and 10 mM urea in PBS) or a more negative charge (NaHCO_3_, Na-gluconate and 10 mM sodium dodecylsulphate (SDS)) in PBS [1].

Radiolabeled compounds were purchased from PerkinElmer (Waltham, MA, USA). For TJ disruption apical application of 150 μM ACA for 3 h [18] or the peroxisome proliferator-activated receptor gamma inhibitors T0070907 (10 μM) or GW9662 (1 μM) for 1 h were used [19].

siRNA knockdown experiments were employed to test the functionality of proteins involved in TJ constitution and proteins of significance for PC translocation. As in earlier studies with CaCo2 cells, we used in each case 78 pmol siRNA scrambled as control and targeted to claudin-1, -2, -4 and -8, Zonula Occludens-1 (ZO1), kindlin-1 and -2, CFTR, AE2 and mucin 1, mucin 2, and mucin 3 [1]. siRNAs were apically applied for 16 h at 37 °C. After washing transport studies were initiated.

### 4.2. siRNA Knockdown Experiments

All sense and antisense probes except for kindlin-1 and kindlin-2 were obtained from Sigma (St. Louis, MO, USA) and are depicted in Table 1. The probes for kindlin-1 and kindlin-2 were a kind gift of Reinhard Faessler, Max Planck Institute of Biochemistry, Munich, Germany. The efficiency of siRNA knockdown was confirmed by Western blotting. Transport characteristics included dependency on time, concentration, temperature and pH and was evaluated as described [1].

### 4.3. Immunoblot Analysis

Cell homogenate samples with 30 μg protein were applied to the gel slots for electrophoretic separation and immunoblotting using a standard protocol [1]. Primary antibodies against the following human proteins were used: claudin 2 (cat. no.: TA347352; 1:500) (Acris Antibodies, Herford, Germany); ZO1 (cat. no.: ab96587; 1:200; Abcam, Cambridge, MA, USA); kindlin-2 (1:500; gift from Reinhard Faessler, Max Planck Institute of Biochemistry, Munich, Germany); mucin 2 (cat. no.: sc-59859; 1:500; Santa Cruz Biotech., Santa Cruz, CA, USA) and CFTR (sc-376683; 1:200; Santa Cruz Biotech.

### 4.4. Statistical Analysis

Statistical analysis was performed using Prism 4.0 software (GraphPad Software Inc., LaJolla, CA, USA). Differences between groups were evaluated using the Mann–Whitney U test. Data are presented as means ± SD and *p* < 0.05 was considered statistically significant.

## 5. Conclusions

The manuscript shows for the first time that phosphatidylcholine can pass by paracellular transport from systemic sources to the apical side of biliary epithelial cells. It passes the TJ barrier driven by an apical negative electrical potential, generated by CFTR and AE2. Mucin 3 is the initial apical acceptor molecule from where PC is handled to secretory mucin 1 and 2 to establish the barrier against the biliary lumen. It protects from high bile acid concentration within the biliary lumen (Figure 11).

## Figures and Tables

**Figure 1 ijms-20-04034-f001:**
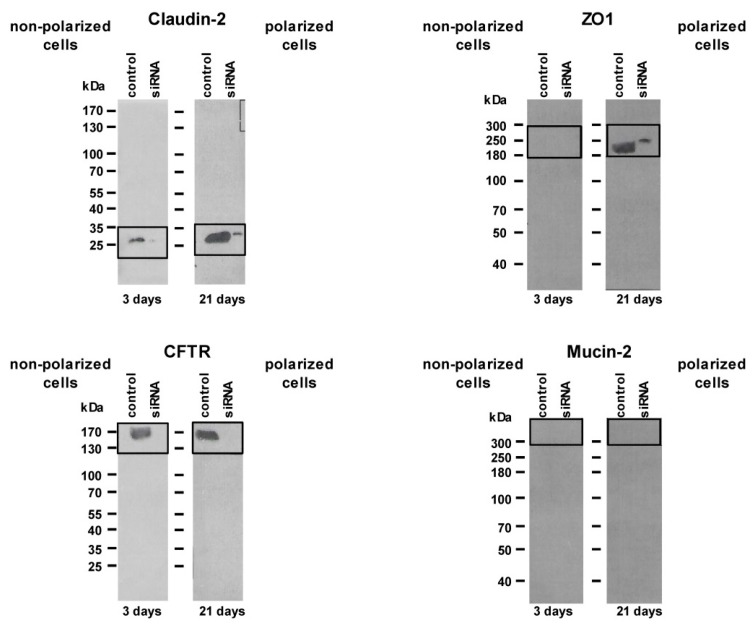
Western blot with non-polarized (three days cultured) and polarized (21 days cultured) MzChA-1 cells with representative proteins of the tight junction (TJ) complex (claudin-2; Zonula Occludens-1, ZO1). Cystic fibrosis transmembrane conductance regulator (CFTR) is a constitutive, non-TJ protein used as housekeeping gene, and mucin 2 as a protein that is not expressed in cholangiocytes. Applied were 30 μg of cell homogenates. The data indicate the enhancement of appearance of the TJ proteins claudin-2 and ZO1 with the polarization of the cells.

**Figure 2 ijms-20-04034-f002:**
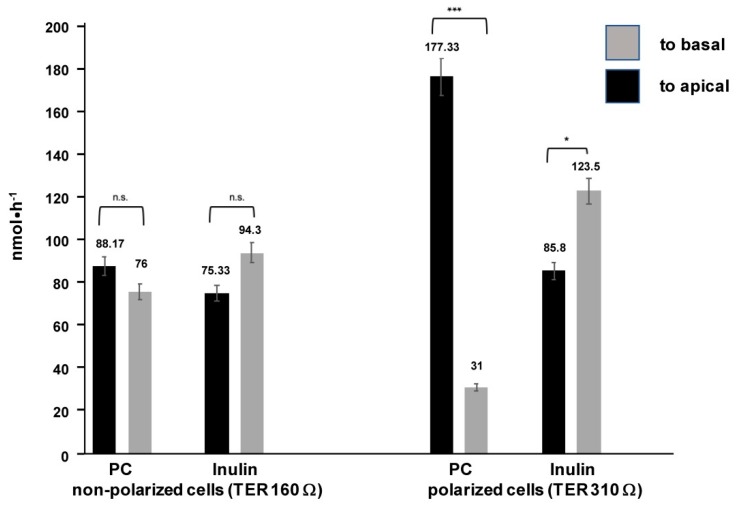
Apical vs. basal transport of phosphatidylcholine (PC) and inulin applied to a transwell culture system of non-polarized (three days cultured) and polarized (21 days cultured) Mz-ChA-1 cells. Means ± SD of *n* = 6; n.s. = not significant, * *p* < 0.05, *** *p* < 0.001. TER, transepithelial resistance.

**Figure 3 ijms-20-04034-f003:**
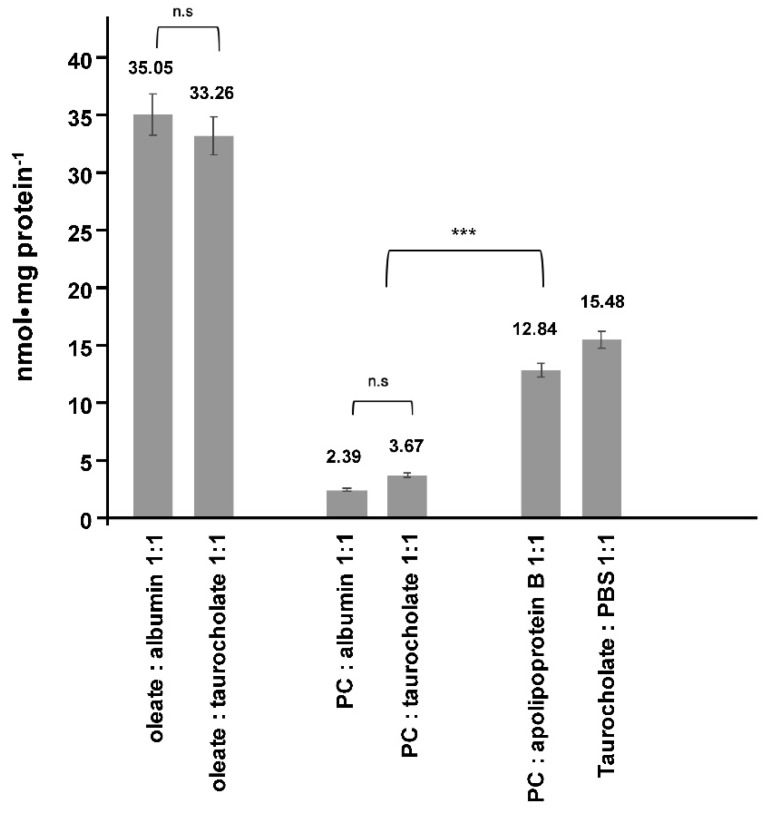
Intracellular accumulation of phosphatidylcholine (PC), fatty acids and taurocholate (TC) in non-polarized Mz-ChA-1 cells. After 1 h incubation of the different substrates with various binding molecules at 37 °C, the amount taken up by the cells was determined after washing the cells three times with 10 mM TC in phosphate-buffered saline (PBS). Means ± SD of *n* = 6; n.s. = not significantly different, *** *p* < 0.001.

**Figure 4 ijms-20-04034-f004:**
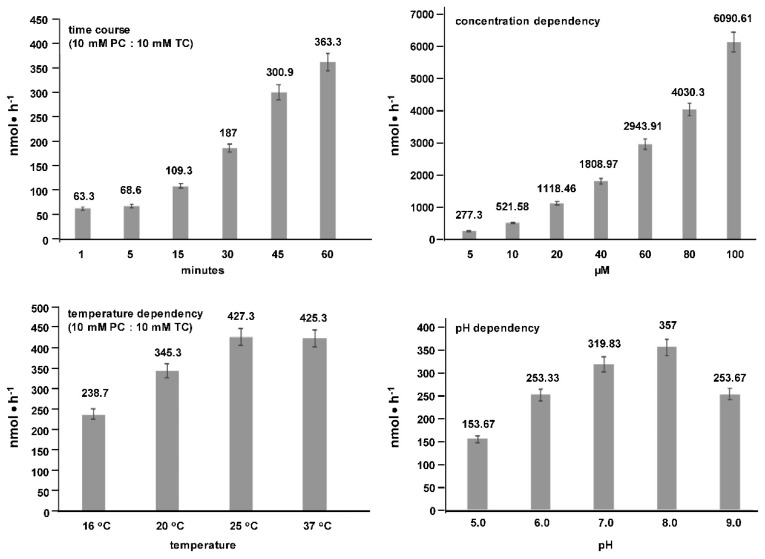
Transport characteristics of phosphatidylcholine (PC) to the apical side of polarized Mz-ChA-1 cells. Apical transport of PC was evaluated after exposure to the basal side. Transport was linear in regard to time and concentration (1 h incubation) with a temperature optimum between 25–37 °C and a pH optimum in the range of pH 7.0 to pH 8.0. Means ± SD of *n* = 6.

**Figure 5 ijms-20-04034-f005:**
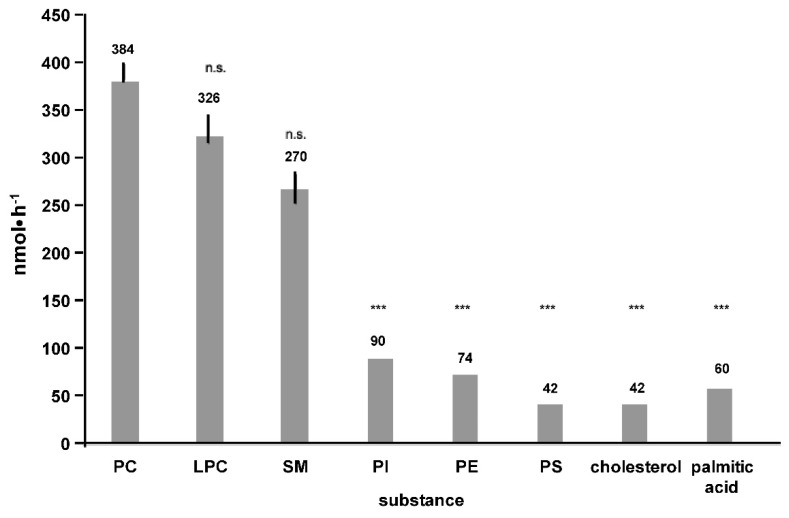
Specificity of apical transport for choline-containing phospholipids in polarized, nine-day cultured Mz-ChA-1 cells. Apical transport was examined for 1 h after basal application of the different substrates at 10 mM phosphatidylcholine (PC) together with 10 mM taurocholate (TC). Means ± SD of *n* = 6. Significances were calculated in relation to PC; n.s. = not significant, *** *p* < 0.001. Abbreviations used are: LPC, lysophosphatidylcholine; SM, sphingomyelin; PI, phosphatidylinositol; PE, phosphatidylethanolamine; PS, phosphatidylserine.

**Figure 6 ijms-20-04034-f006:**
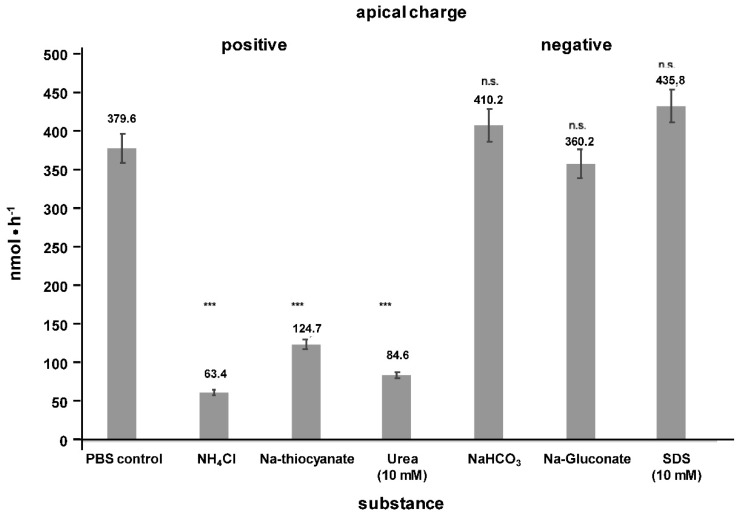
Ionic driving forces for apical translocation of phosphatidylcholine (PC) in polarized Mz-ChA-1 cells. Apical transport of basally applied 10 mM PC: 10 mM taurocholate was examined in polarized (21 days cultured) Mz-ChA-1 cells apically equilibrated for 1 h with indicated salts and substrates (130 mM) generating apical positive or negative charge. Means ± SD of *n* = 6. Significances were calculated in relation to the phosphate-buffered saline (PBS) control; n.s. = not significant, *** *p* < 0.001.

**Figure 7 ijms-20-04034-f007:**
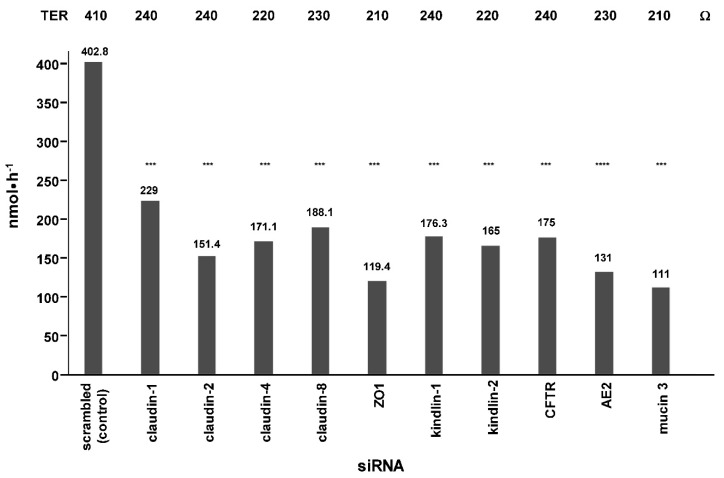
Effect of siRNA suppression of proteins involved in apical translocation of phosphatidylcholine (PC). Apical transport of PC after basal application of 10 mM PC: 10 mM taurocholate was examined after siRNA knockdown of indicated proteins which are all involved in tight junction-mediated luminal secretion of PC. Means ± SD of *n* = 6. Significances were calculated in relation to the control pretreated with scrambled siRNA; n.s. = not significant, *** *p* < 0.001. ZO1, Zonula Occludens-1; CFTR, cystic fibrosis transmembrane conductance regulator; AE2, anion exchange protein 2; TER, transepithelial resistance.

**Figure 8 ijms-20-04034-f008:**
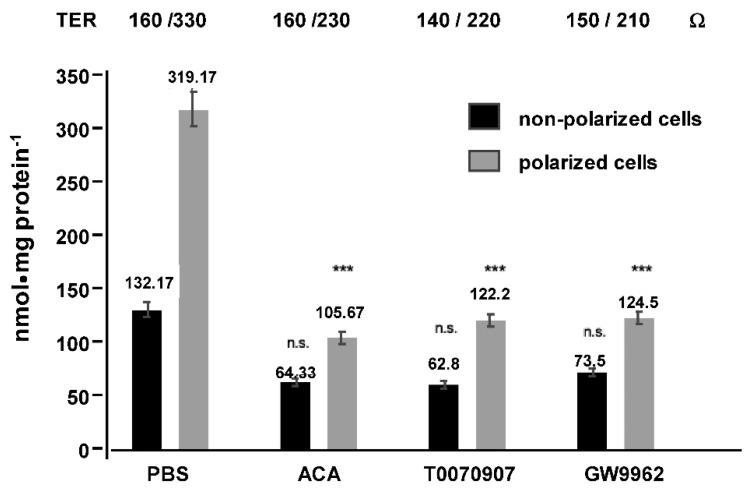
Effect of disruption of tight junctions (TJ) by acetaldehyde (ACA) or the indicated peroxisome proliferator-activated receptor γ (PPARγ) inhibitors on apical transport of phosphatidylcholine (PC). TJ were disrupted by exposure to ACA vapor or incubation of upper transwell chambers with the PPARγ inhibitors. Then apical transport of 10 mM PC: 10 mM taurocholate for 1 h was registered. Means ± SD of *n* = 6. Significances were calculated separately for non-polarized and polarized Mz-ChA-1 cells and compared to phosphate-buffered saline (PBS)-treated controls; n.s. = not significant, *** *p* < 0.001.

**Figure 9 ijms-20-04034-f009:**
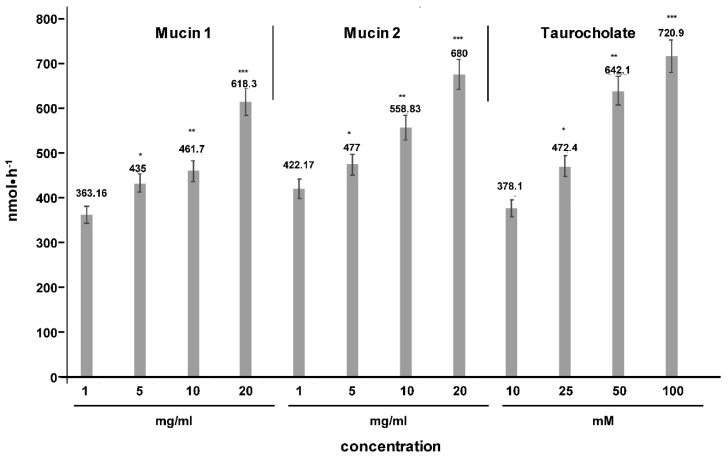
Enhancement of apical phosphatidylcholine (PC) transport by application of increasing concentrations of secretory mucin 1, mucin 2, and taurocholate (TC) as luminal acceptor substances. Apical transport of 10 mM PC: 10 mM TC with polarized Mz-ChA-1 cells was determined as a function of increasing concentrations of mucin 1, mucin 2 and TC in the upper culture system. Means ± SD of *n* = 6. Significances were calculated in relation to the lowest concentration of each substance added. * *p* < 0.05; ** *p* < 0.01; *** *p* < 0.001.

**Figure 10 ijms-20-04034-f010:**
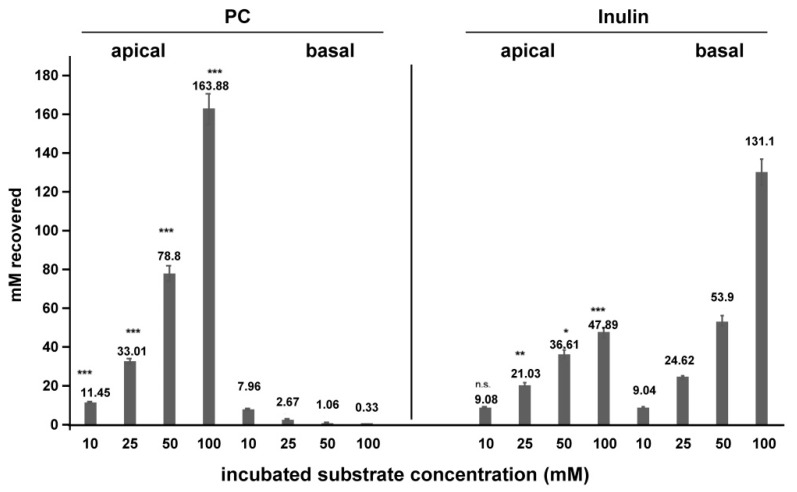
Equilibrium distribution of phosphatidylcholine (PC) and inulin when applied in increasing concentrations to upper and lower compartment of the transwell culture system. Increasing concentrations up to 100 mM of PC and inulin were added to the apical and basal compartments. Over 1 h observation, vectoral transport of PC was completely directed apically with decreasing PC at the luminal side (less than 1%). For inulin there was a significantly higher basal accumulation starting at 25 mM incubated. The process appeared to be less effective, at least at lower concentrations, as for the transport of PC to the apical side. Means ± SD of *n* = 6. Significances were calculated for apical versus basal transport for each in relation to the lowest concentration of each substrate concentration. n.s. = not significant; * *p* < 0.05; ** *p* < 0.01; *** *p* < 0.001.

**Figure 11 ijms-20-04034-f011:**
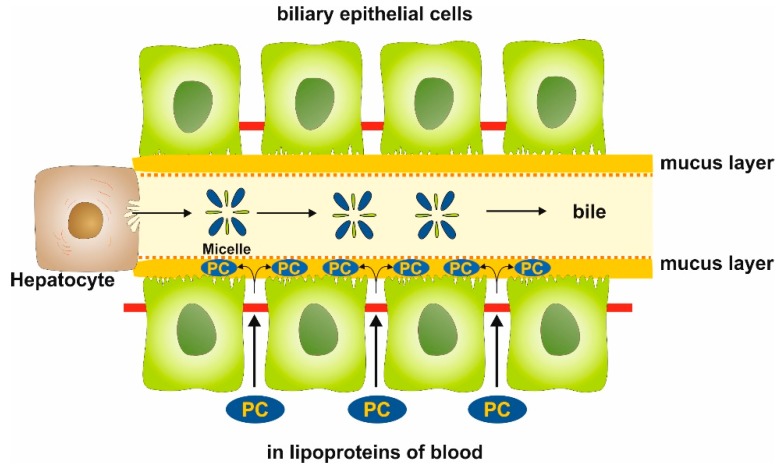
Scheme illustrating the paracellular transport of phosphatidylcholine (PC) across the tight junction barrier to the mucus layer at the apical side of biliary epithelium. The transport is driven by a negative electrical gradient with consequent binding to membrane-localized mucin 3 and an equilibrated shift to secretory mucin 2.

**Table 1 ijms-20-04034-t001:** siRNAs used in knockdown Experiments.

siRNA	Sense Oligo	Antisense Oligo
scrambled	5′-gaugggaccuggccaguga-3′[dT][dT]	5′-ucacuggccaggucccauc-3′[dT][dT]
claudin-1	5′-cagucaaugccagguacga-3′[dT][dT]	5′-ucguaccuggcauugacug-3′[dT][dT]
claudin-2	5′-gacacuaccacuggaucgu-3′[dT][dT]	5′-acgauccagugguaguguc-3′[dT][dT]
claudin-4	5′-gaccaucugggagggccua-3′[dT][dT]	5′-uaggcccucccagaugguc-3′[dT][dT]
claudin-8	5′gguucaagcaucuacucuu-3′[dT][dT]	5′-aagaguagaugcuugaacc-3′[dT][dT]
mucin 2	5′-gcaacauuaccgucugcaa-3′[dT][dT]	5′-uugcagacgguaauguugc-3′[dT][dT]
mucin 3	5′-ccaaacuacucuuacuaca-3′[dT][dT]	5′-uguaguaagaguaguuugg-3′[dT][dT]
CFTR	5′-gaacacauaccuucgauau-3′[dT][dT]	5′-auaucgaagguauguguuc-3′[dT][dT]
AE2	5′-gagaucuucgccuucuuga-3′[dT][dT]	5′-ucaagaaggcgaagaucuc-3′[dT][dT]
ZO1	5′-gagaugaacgggcuacgcu-3′[dT][dT]	5′-agcguagcccguucaucuc-3′[dT][dT]
kindlin-1	5′-ggacauuacugauaucccu-3′[dT][dT]	5′-agggauaucaguaaugucc-3′[dT][dT]
kindlin-2	5′-gugugaauagaaauacugu-3′[dT][dT]	5′-acaguauuucuauucacac-3′[dT][dT]

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
