# Peer review of "Phosphatidylcholine Passes by Paracellular Transport to the Apical Side of the Polarized Biliary Tumor Cell Line Mz-ChA-1"

_ijms, 2019, doi:10.3390/ijms20164034_

Round 1
Reviewer 1 Report
The present manuscript shows that phosphatidylcholine (PC) can pass by a paracellular transport from systemic sources to the apical side of cholangiocytes, thanks to an apical negative electrical potential generated by CFTR and AE2, with consequent binding to membrane-localized mucin 3 and an equilibrated shift to secretory mucin 2. It is very interesting but there are some points to elucidate and to enrich:
- The Authors, in the Abstract, in the Introduction and in the Discussion sections, have mentioned the primary sclerosing cholangitis (PSC) as a disease in which PC translocation into mucus via a paracellular transport across the apical/lateral tight junction (TJ) barrier may have implications for its pathogenesis. Therefore my question is: why did they use a cholangiocarcinoma (CCA) cell line, in particular a typical extrahepatic CCA cell line? The Authors need to clarify this point.
- Related to the previous point, I am not completely agree with the sentence “These cells share indeed the characteristics of physiologic biliary epithelial cells” (line 59). In fact, we have the reference but the sentence remains too general, the Authors should explain better which are the physiologic characteristics of Mz-ChA-1. For that reason, I suggest to add a non-malignant cholangiocyte cell line, also to compare the important data obtained.
- It should be really interesting also to add some images of PC movement across the polarized and unpolarized cholangiocytes.
- In the Material and Methods section, the titles 4.1 and 4.3 are the same, they could change and specify one or put together just in one paragraph.
Reviewer 2 Report
The paper by Stremmel et al., analyses the paracellular translocation of PC across the polarized biliary epithelial tumor cell line Mz-ChA-1.
The authors found that the transport is unidirectional to the PC apical side, mediated by tight Junctions, driven by negative apical charge generated by CFTR and AE2, and stimulated by apical application of secretory mucins.
The authors evaluated the transport characteristics of PC to the apical side in regard to time, concentration, temperature and pH.
Conclusions about the fact that PC is bound to mucin 3 is not clearly documented and supported by experiments.
Line 44-45 add a reference
In the references, there is an abundance of self-citation (7 on 16)
Specific comments:
Fig 1 molecular weight in the western blot is expressed as KD , but are Daltons (ex ZO1 is 250 KD and not 250000 kd). Furthermore, there is not shown a protein housekeeping or a densitometric analysis.
Round 2
Reviewer 1 Report
In my opinion, the present revised version could be suitable for
publication in the International Journal of Molecular Sciences.
Author Response
Dear reviewer,
many thanks for your evaluation.